# A facile way to construct sensor array library via supramolecular chemistry for discriminating complex systems

Jia-Hong Tian[1,2], Xin-Yue Hu[1,2], Zong-Ying Hu[1], Han-Wen Tian[1], Juan-Juan Li[1], Yu-Chen Pan[1], Hua-Bin Li[1] & Dong-Sheng Guo [1][✉]

Differential sensing, which discriminates analytes via pattern recognition by sensor arrays, plays an important role in our understanding of many chemical and biological systems. However, it remains challenging to develop new methods to build a sensor unit library without incurring a high workload of synthesis. Herein, we propose a supramolecular approach to construct a sensor unit library by taking full advantage of recognition and assembly. Ten sensor arrays are developed by replacing the building block combinations, adjusting the ratio between system components, and changing the environment. Using proteins as model analytes, we examine the discriminative abilities of these supramolecular sensor arrays. Then the practical applicability for discriminating complex analytes is further demonstrated using honey as an example. This sensor array construction strategy is simple, tunable, and capable of developing many sensor units with as few syntheses as possible.

[1] College of Chemistry, Key Laboratory of Functional Polymer Materials (Ministry of Education), State Key Laboratory of Elemento-Organic Chemistry, Nankai University, Tianjin 300071, China. [2]These authors contributed equally: Jia-Hong Tian, Xin-Yue Hu. [✉]email: dshguo@nankai.edu.cn

nspired by the human tongue and nose, differential sensing systems (alternately called chemical nose or E-nose) have been developed to enable the discrimination and identification of analytes with similar structures or complex mixtures of unknown structures/components[1,2]. Compared with single sensors that depend on selectivity for a particular analyte, sensor arrays take into account the simultaneous cross-reactive interactions of multiple analytes and sensor units, in order to create a unique pattern or fingerprint of each analyte[3–5]. Nowadays, sensor arrays are widely used in industry and research[6–10] to detect analytes related to human health[11–14], environment[15–17], quality control[18–20], and others. To ensure satisfactory differentiating performance, it is necessary to combine a sufficient number of sensor units to form the sensor array. In general, more sensor units mean a better differentiating index, because they can potentially provide more information between the sensor array and the analytes. For example, in order for the human olfactory system to differentiate among all possible volatile compounds and the huge number of their combinations, it consists of over 1000 active and highly cross-reactive receptors[21]. Therefore, a key task of creating an effective differential sensing system is to build a library of sensor units. Chemical synthesis is a powerful approach to obtain new sensor units. However, building many sensor units only through covalent chemistry is laborious and time-consuming because of the additional synthesis and purification work, consumes extra chemicals, as well as involves considerable research and development costs. Therefore, there is an urgent need to develop new, simpler methods to construct sensor unit libraries.

In addition to covalent chemistry, supramolecular chemistry represents another elegant approach to construct multi-functional materials[22–25]. The dynamic nature of supramolecular chemistry simplifies the construction of sensor units in the array. For instance, the indicator displacement assay (IDA), in which a competitive analyte is introduced to a dye/receptor system and the displacement of dye by analyte modulates the optical signal, represents a popular supramolecular strategy to construct sensor array[6,26]. If the sensor units are constructed by compounds possessing recognition and assembly properties, then there will be almost unparalleled diversity in these sensor units because the supramolecular components and their ratios can be easily tuned without further synthesis. For instance, consider two receptors that can coassemble with each other and two dyes. If we assume that one dye can only be complexed with one receptor, then by using five different coassembly ratios and five receptor/dye ratios, a 50-sensor unit library can be built as shown in Fig. 1. As the recognition properties among the receptors, dyes, and analytes are different, the analytes displace different amounts of dyes in the sensor units. Moreover, during titration, the nonlinear relationship between the output signal and the component concentration can be exploited to construct sensor units by simply adjusting the receptor/dye ratios. This nonlinear relationship has ever been used in supramolecular encryption[27,28], but is overlooked in differential sensing. Such sensor units generate different signals for each analyte due to complex supramolecular equilibria even the binding affinities among receptor, dye, and analyte stay the same. Thus, each analyte will have a fingerprint constructed by 50 different output signals, which provides abundant information for discrimination. This idealized model clearly shows the power of supramolecular chemistry. To date, many sensor arrays based on supramolecular systems have been developed[29–41]. However, most of the research has focused on their recognition properties. The assembly property, which is the other powerful weapon from supramolecular chemistry, has not been given adequate attention[42,43]. Supramolecular chemistry can be a more effective strategy for constructing a sensor unit library by simultaneously utilizing both molecular recognition and assembly.

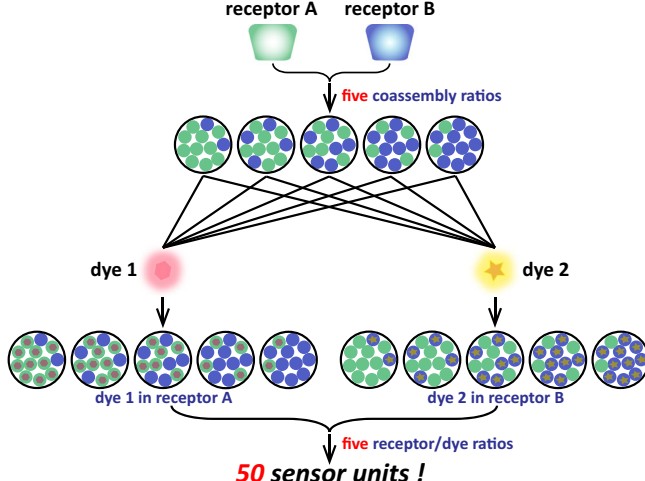

**Fig. 1 Formation of 50 sensor units utilizing two receptors and two dyes.** First, changing the coassembly ratio produces five different coassembled receptors. Then, the two dyes are separately introduced into this system to give 10 sensor units. Finally, by using five receptor/dye ratios in each sensor unit, 50 sensor units are prepared. The supramolecular sensor arrays can discriminate analytes based on the different binding affinities between receptors/analytes and receptors/dyes, as well as the nonlinear relationship between the fluorescence intensity and the analyte concentrations.

In this work, we fully utilize both the recognition and assembly properties of receptors to develop a supramolecular sensor array library. Amphiphilic macrocyclic hosts, which are considered surfactants with host-guest recognition sites, are artificial receptors that possess recognition as well as assembly properties. They are good candidates for developing supramolecular sensor arrays. As a proof of concept, we coassemble various amphiphilic calixarenes (CAs) with cyclodextrin (CD) to construct different receptors, and the output optical signals are produced from the competitive displacement of dyes from the receptor by the analytes, which allows the recognition events to be observed. Considering their heteromultivalent binding abilities that can provide multiple and diverse binding sites to the analytes, these sensor arrays may be suitable for identifying biomacromolecules or complex mixtures. In this study, we use proteins as model analytes to test the discriminative properties of the supramolecular sensor arrays constructed using the strategies mentioned above. Next, honey as a representative complex mixture is successfully discriminated.

## Results

**Building blocks of sensor arrays and formation of supramolecular sensor units.** An efficient sensor unit should possess good recognition ability. Here, we chose nine different macrocycles, namely guanidinium calix[n]arenes (GCnAs, $n = 4$ or 5)[44,45], quaternary ammonium calix[n]arenes (QCnAs, $n = 4$ or 5), sulfonated calix[n]arenes (SCnAs, $n = 4, 5, 6, 8$)[46,47] and amphiphilic $\beta$-CD[48] (Fig. 2 and Supplementary Figs. 1–9, 11–22) as the building blocks of coassembled receptors, because of their excellent abilities to recognize different guest molecules[37,45,49–53]. The receptors were prepared by hydrating a mixture of CA (CA = GCnA, QCnA, or SCnA) and CD in water under sonication at 80 °C for 3 h. The CA-CD receptors showed a hydrated diameter from 30 nm to 50 nm, and their surface potential was ~50 mV for GCnA-CDs and QCnA-CDs and −50 mV for SCnA-CDs, according to dynamic light scattering and zeta potential results, respectively (Supplementary Fig. 24). These receptors

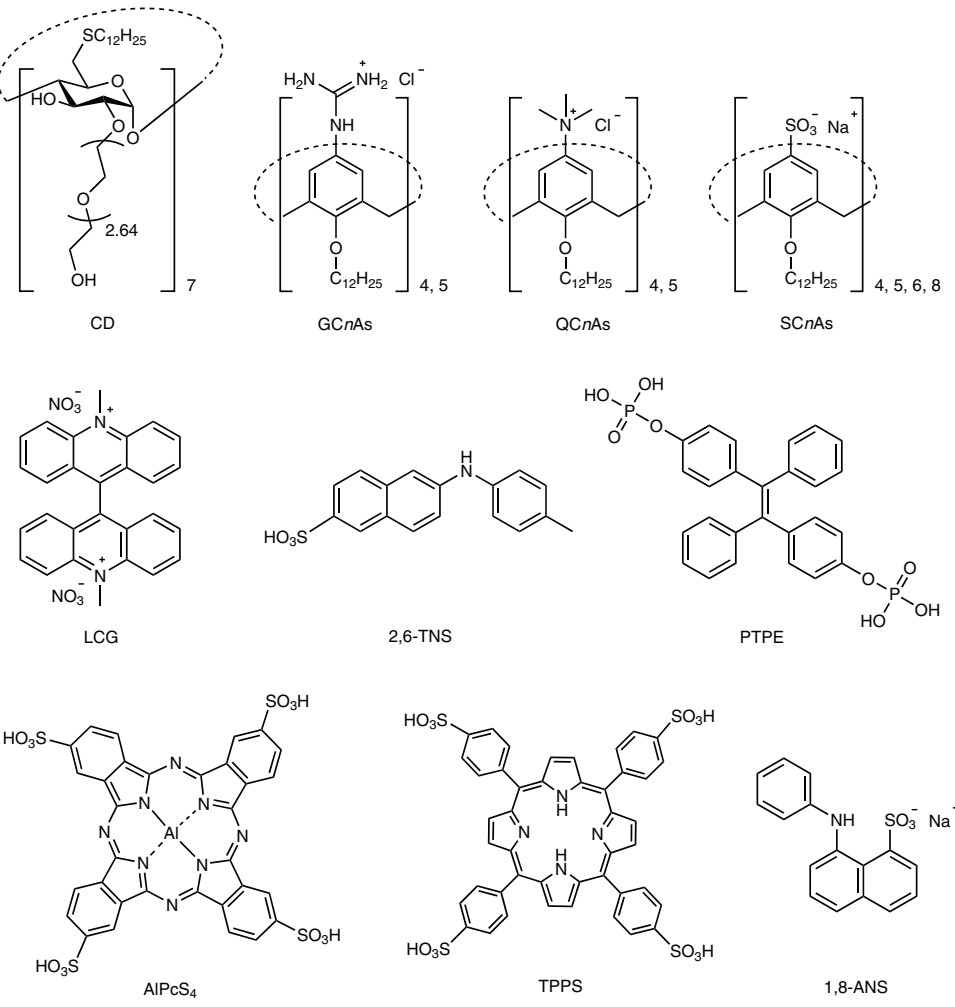

**Fig. 2 Chemical structures of CD, CAs, and dyes.** Chemical structures of the employed CA (GC*n*As, QC*n*As, and SC*n*As) and CD hosts, and the reporter dyes (LCG, 2,6-TNS, PTPE, AlPcS$_4$, TPPS, and 1,8-ANS) in this work.

have multiple and diverse recognition sites on their surface that can fit different binding sites on proteins or other complex systems[51,54,55].

The other key requirement of the sensor unit is producing an easily observable signal that reflects the recognition event between the sensor and analyte. In this case, the output signals were generated by conducting IDA[6,56,57]. An indicator is bound to a receptor to create the reporter pair. Subsequent introduction of the analyte causes displacement of the indicator from the receptor to produce a measurable output signal. IDA is compatible with differential sensing, because an array can be constructed by combining multiple receptors and indicators without additional synthetic efforts[6,58]. In this study, the indicators are six fluorescent dyes (Fig. 2) that can be complexed by the CA-CD receptors: Al(III) phthalocyanine chloride tetrasulfonic acid (AlPcS$_4$), 5,10,15,20-tetrakis(4-sulfonatophenyl)porphyrin (TPPS), sodium 8-anilino-1-naphthalenesulfonate (1,8-ANS), phosphated tetraphenylethylene (PTPE), 2-(*p*-toluidinyl)naphthalene-6-sulfonic acid (2,6-TNS), and lucigenin (LCG). AlPcS$_4$ and TPPS are indicators for GC*n*As and QC*n*As with complexation-induced quenching. 1,8-ANS, PTPE, and 2,6-TNS are indicators for GC*n*As and QC*n*As with complexation-induced enhancement. LCG is an indicator for SC*n*As with complexation-induced quenching (Supplementary Fig. 25). A number of sensor units can be generated by choosing different CAs and dyes, changing the ratio of CA to CD, tuning the ratio of dyes to CAs, and varying the

environmental variables (such as pH of the sensor array). These sensor units were then used to create different sensor arrays, and their ability to distinguish complex systems was examined.

**Sensor array based on different reporter pairs**. The first type of sensor array was constructed by changing both the receptors and the corresponding dyes (Supplementary Fig. 27a). In order to explain the discrimination principle of the sensor array more clearly, we first present a theoretical model based on host–guest recognition[59]. The discriminative ability of this sensor array comes from different binding strengths between the receptors/ dyes and the receptors/analytes. Three receptors and three dyes were complexed to form the sensor units. We assumed that each receptor–dye pair has a given binding constant, which was used to calculate the concentrations of complexed and free dye molecules in the system at given initial receptors and dye concentrations (Supplementary Table 3). Subsequently, three target analytes were added to the sensor units. Each analyte has a specific binding constant with each receptor. The addition of the analyte causes release of the dye from the receptor. The concentration of the free dye at this time was also calculated (Supplementary Table 3). If we assume that the dye concentrations remain in the linear range of fluorescence intensity vs. concentration, then the ratio of fluorescence intensity before ($I_0$) and after adding analytes ($I$) can be predicted (Supplementary Fig. 27b–d and Supplementary Table 3). Supplementary Table 4

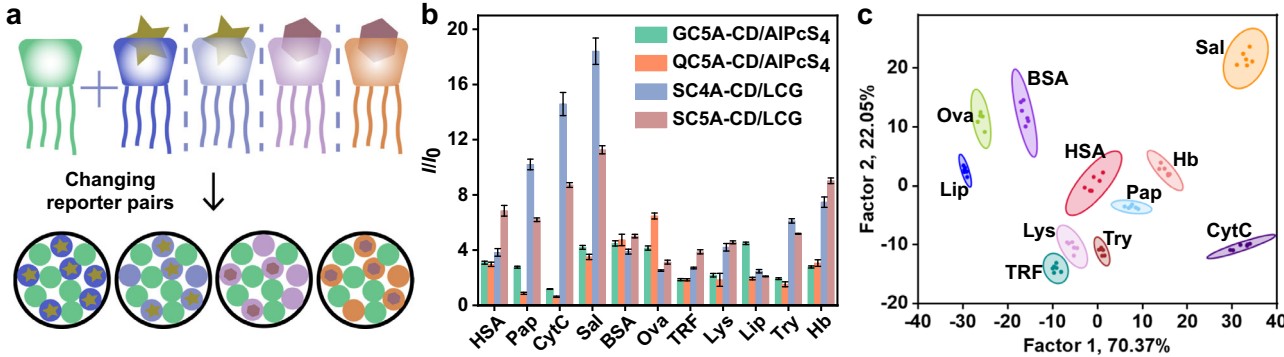

**Fig. 3 Construction and experimental results of sensor array based on different reporter pairs. a** Schematic diagram of **SA1**. Pattern recognition of proteins using **SA1** ([CA] = [CD] = 1.0 μM, [dye] = 1.0 μM). **b** Fluorescence response patterns of **SA1** against various proteins (23.8 μg/mL for each protein). The order of colored symbols from top to bottom in the legend corresponds to the column order of histogram from left to right. **c** Canonical score plot for the two factors of simplified fluorescence response patterns obtained from LDA with 95% confidence ellipses. All experiments were performed in water (pH = 6.53) and pH only slightly changed after the addition of proteins (Supplementary Table 2). The fluorescence spectra of dyes were recorded and kept almost unchanged in the presence of NaClO$_4$ (up to 27.5 mM, Supplementary Fig. 26) in order to exclude the effect of ionic strength. Error bars in **b** represent mean ± s.d. ($n$ = 6 independent experiments).

shows the data used to simulate the sensor array for discriminating the three different analytes. According to the simulation, $I/I_0$ depends on the affinity between the receptor/dye and the receptor/analyte. A lower binding constant of the former and a higher binding constant of the latter will lead to a higher ratio of the replaced dye concentration to the initial free dye concentration (a larger $I/I_0$ value), and vice versa. These $I/I_0$ values form the fingerprint of each analyte. Here, we randomly assigned some values around the calculated fluorescence response value to simulate repeated experiments (Supplementary Table 4 and Supplementary Fig. 27e). Linear discriminant analysis (LDA) is a statistical method that minimizes intraclass variance and maximizes interclass variance to differentiate between response patterns. LDA analysis of the simulated data (Supplementary Fig. 27f) shows that the sensor array could identify and classify the considered analytes.

Based on the theoretical modeling, an actual sensor array was constructed from the CA-CD assemblies. This sensor array (**SA1**) was composed of four sensor units: GC5A-CD/AlPcS$_4$, QC5A-CD/AlPcS$_4$, SC4A-CD/LCG, and SC5A-CD/LCG (Fig. 3a). The complexation of CA-CD leads to fluorescence super quenching that can provide a lower initial fluorescence intensity[45], making the system more sensitive for the fluorescence switch-on sensing after adding analytes. Thirteen proteins were used as analytes: human hemoglobin (Hb), transferrin (TRF), cytochrome C (CytC), lysozyme (Lys), myoglobin (Mb), lipase (Lip), trypsin (Try), human serum albumin (HSA), bovine serum albumin (BSA), bovine hemoglobin (BHb), chicken egg white albumin (Ova), papain (Pap), and salmine sulfate (Sal) (Supplementary Table 1). The proteins were added to the four-element sensor array, and the fluorescence changes at the maximum emission wavelengths were recorded in six repetitions using a fluorescence spectrometer. The ratio of fluorescence intensity after and before protein addition ($I/I_0$) was used as the signal response value. Figure 3b shows that the four sensor units provided completely different fluorescence response patterns for different proteins. For instance, BSA produced very similar fluorescence responses from the four sensor units. In contrast, the fluorescence signals for Sal generated by sensor units composed of negatively charged calixarenes are significantly higher than those of positively charged calixarenes, which may be attributed to the strong affinities between Sal and negatively charged calixarenes. Each sensor unit provides information about the proteins, and the combined information gives an exclusive recognition pattern for each protein. Using

these fingerprint-like fluorescence responses, the proteins can be effectively discriminated by LDA analysis. Out of the 13 proteins, 11 were successfully classified. The two most significant LDA factors (F1 = 70.37% and F2 = 22.05%) were used to generate a two-dimensional score plot with 95% confidence ellipses, and all 66 points (11 proteins × 6 replicates) were well clustered into 11 distinct groups without any overlap (Fig. 3c). The correct classification and jackknifed classification of the LDA analysis indicated 100% accuracy in differentiating the patterns.

Encouraged by the excellent discrimination using multiple reporter pairs, we considered a second sensor array (**SA2**) made of four positively charged coassemblies (GC4A-CD, GC5A-CD, QC4A-CD, and QC5A-CD) and the dye AlPcS$_4$. **SA2** was simpler than **SA1** since the dye was fixed. The different binding affinities between the receptors and AlPcS$_4$ and between the receptors and proteins still play decisive roles in the sensing events. The results showed that **SA2** identified 9 of the 13 proteins (Supplementary Fig. 28). This strongly proves that the supramolecular sensor arrays constructed by changing the reporter pairs are capable of discriminating biological macromolecules such as proteins.

**Sensor array based on dye replacement**. In the second type of sensor array we proposed, the coassembly is fixed and different sensor units are developed by changing the dyes (Fig. 4a). In our case, the synthesis workload mainly consisted of the syntheses of receptors, while the dyes were commercially available. Owing to the recognition compatibility of macrocycles[60–63], it is not difficult to find enough kinds of dyes to construct the sensor array. First, simulation was carried out using a sensor array model of one receptor and three dyes. Unlike the previously discussed sensor arrays (**SA1** and **SA2**), the single receptor here means that the binding affinities between the receptor and the analytes remain the same, and the discriminative ability comes from the different binding capabilities of the receptor and dyes. The data used to simulate the sensor array are presented in Supplementary Tables 5 and 6. The simulated fluorescence signals allow visual distinction of the analytes on the LDA plot (Supplementary Fig. 29). These theoretical results prove the possibility of designing a sensor array based on replacing the dyes.

The model provides the theoretical possibility, and the point of experiment is to show that it truly works. We performed experiments using only one coassembly (GC5A-CD) and four different dyes (AlPcS$_4$, TPPS, 1,8-ANS, and PTPE) to construct the four-element sensor array **SA3**. As shown in Fig. 4b, a distinct

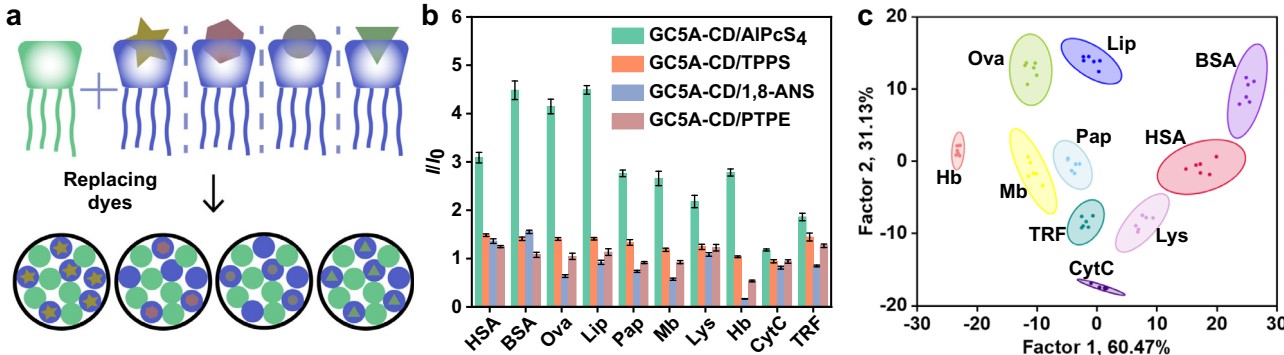

**Fig. 4 Construction and experimental results of sensor array based on dye replacement. a** Schematic diagram of **SA3**. Pattern recognition of proteins using **SA3** ([GC5A] = [CD] = 1.0 µM, [dye] = 1.0 µM). **b** Fluorescence response patterns of **SA3** against various proteins (23.8 µg/mL for each protein). The order of colored symbols from top to bottom in the legend corresponds to the column order of histogram from left to right. **c** Canonical score plot for the two factors of simplified fluorescence response patterns obtained from LDA with 95% confidence ellipses. Error bars in **b** represent mean ± s.d. (n = 6 independent experiments).

recognition pattern was observed for each protein. **SA3** demonstrated good classification performance on the canonical score plot. From LDA analysis, clusters representing ten proteins were located in different areas, revealing a 100% classification accuracy (Fig. 4c). The discriminative effect of **SA3** (constructed by changing the dyes) was even better than that of **SA2** (changing the receptors) to some extent. This is a little surprising since **SA3** does not have different binding interactions between the receptors and the same analyte. The likely reason is that the interactions between the dyes and analytes are also important in the classification. These interactions can be observed from the fluorescence changes. 1,8-ANS and PTPE showed enhanced fluorescence upon complexation with GC5A-CD, and so the dye displacement caused the signal to drop. Proteins such as Hb resulted in a decrease in the fluorescence signal of GC5A-CD/1,8-ANS and GC5A-CD/PTPE sensor units. However, some proteins such as HSA and BSA increased the fluorescence signal of GC5A-CD/1,8-ANS. This is because those proteins also interact with these dyes to enhance the fluorescence[64,65]. Such multiple interaction events enrich the cross-activity of the supramolecular sensor array, thereby favoring the discrimination[41,66,67]. To demonstrate that the success of building sensor arrays by dye replacement can be generalized to other coassemblies, QC5A-CD, with four dyes, AlPcS₄, TPPS, 1,8-ANS, and 2,6-TNS, were used to construct an additional four-element sensor array (**SA4**). As shown in Supplementary Fig. 30, the nine proteins were successfully identified.

**Sensor array based on adjusting the coassembly ratio.** Next, we constructed a sensor array by changing the ratio between the two receptor components used in the coassembly (Fig. 5a). Because of the dynamic reversibility of self-assembly, the CA-CD coassembly can be flexibly adjusted by simply changing the ratio of the two macrocycles. By fixing one macrocycle's concentration and varying the other, a number of sensor units can be obtained without synthesizing more receptors. Meanwhile, fixing the number of dyes to one also avoids purchasing more chemicals. The discriminative ability of this sensor array is due to the different binding affinities between the receptors and analytes. Changing the ratio in the coassembly affects the number, ratio, and distribution of the binding sites on its surface, which will then influence the binding affinities with the same analyte. Supplementary Table 7 shows the theoretically predicted recognition using the sensor array based on adjusting the coassembly ratio. In the simulated array, the ratios between the two components were 2:1, 1:1, and 1:3; the concentration of

the first macrocycle was fixed, and the dye concentration was consistent with the changed component. Similar to previously discussed strategies, our simulation shows that this sensor array based on adjusting the coassembly ratio also distinguish the analytes through LDA (Supplementary Table 8 and Supplementary Fig. 31).

The actually constructed sensor array (**SA5**) used GC5A-CD and AlPcS₄. The concentration of CD was fixed, and the [CA]/[CD] ratio was 1:2, 1:1, 2:1, and 3:1. The concentration of AlPcS₄ was equal to that of GC5A. As shown in Fig. 5b, each coassembly had a different fluorescence signal in response to the same protein, and the combination of four coassemblies would provide the fingerprint for each protein. Among the 13 proteins, 8 could be distinguished by LDA analysis (Fig. 5c). SC4A-CD and LCG were used to further validate the strategy of adjusting the coassembly ratio. The [SC4A]/[CD] ratios used in this sensor array (**SA6**) were 1:2, 1:1, 2:1, and 3:1, respectively. The results showed that **SA6** identified 9 of the 13 proteins (Supplementary Fig. 32).

**Sensor array based on adjusting reporter pair ratio.** Our designed sensor units consist of three parts: CA, CD, and the dye. Next, we built sensor units by adjusting the ratio between the receptor and the dye (Fig. 6a). In previously discussed examples **SA1**–**SA6**, the discriminative abilities come from the different binding affinities among the receptors, dyes, and analytes. In contrast, for the current case the discriminative ability mainly comes from the nonlinear relationship between the fluorescence signal and analyte concentration in the IDA process. In the theoretical model in Fig. 6b, the red line is the titration curve of the receptor and dye, and so $P_0$ and $Q_0$ have different receptor-dye ratios. The blue and green lines represent the competitive titration curves starting at points $P_0$ and $Q_0$, and after adding the same amount of analyte the fluorescence intensities recovered to P and Q, respectively. The changes in fluorescence ($\Delta I_P$ and $\Delta I_Q$) are not equal. Therefore, the sensor array constructed by changing the reporter pair ratios can also provide a recognition pattern for each analyte, although all the receptors have the same binding affinity with a given analyte. The simulated data from the model are presented in Supplementary Table 9. Indeed, the reporter pair with a fixed binding constant showed different fluorescence intensities (free dye concentrations) under the three receptor-dye ratios, which resulted in different initial signal $I_0$. Adding various analytes at the same concentration caused different fluorescence recoveries as the unique response for each analyte (Fig. 6c). As shown in

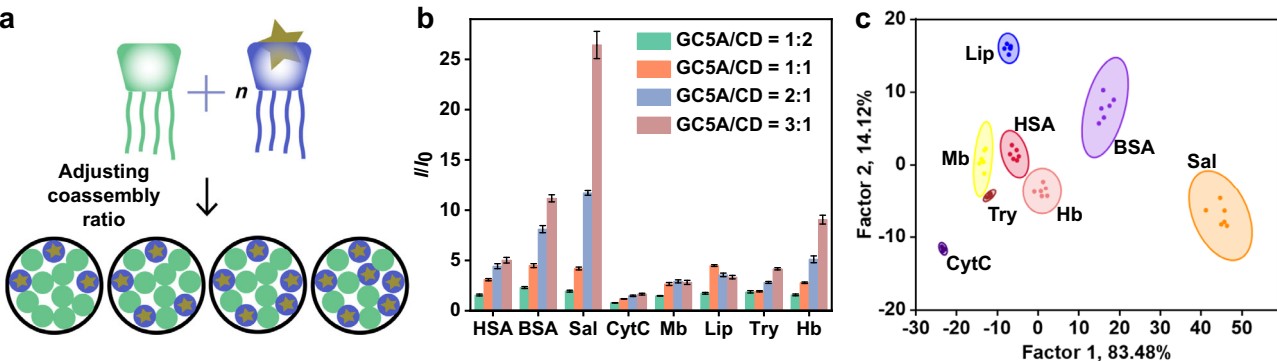

**Fig. 5 Construction and experimental results of sensor array based on adjusting coassembly ratio. a** Schematic diagram of **SA5**. Pattern recognition of proteins using **SA5** ([GC5A]/[CD] = 0.5/1.0 μM, 1.0/1.0 μM, 2.0/1.0 μM, and 3.0/1.0 μM, [AlPcS$_4$] = [GC5A]). **b** Fluorescence response patterns of **SA5** against various proteins (23.8 μg/mL for each protein). The order of colored symbols from top to bottom in the legend corresponds to the column order of histogram from left to right. **c** Canonical score plot for the two factors of simplified fluorescence response patterns obtained from LDA with 95% confidence ellipses. Error bars in **b** represent mean ± s.d. (*n* = 6 independent experiments).

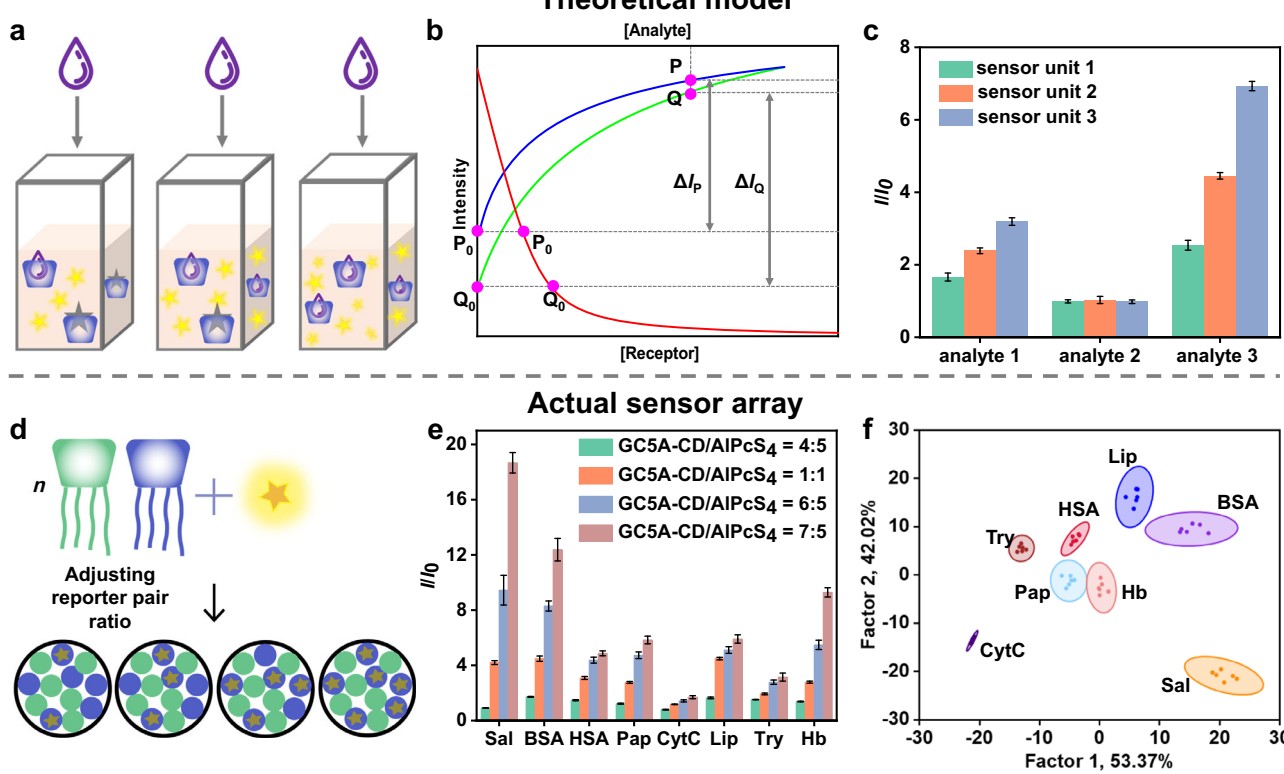

**Fig. 6 Construction, theoretically model, and experimental results of sensor array based on adjusting reporter pair ratio. a** Schematic diagram of the theoretical model of sensor array formed by adjusting the reporter pair ratios. **b** Schematic simulated titration curves for direct binding and competitive binding. **c** Fluorescence response patterns of the simulated sensor array. **d** Schematic diagram of **SA7**. Pattern recognition of proteins using **SA7** ([GC5A] = [CD] = 0.8 μM, 1.0 μM, 1.2 μM, and 1.4 μM, [AlPcS$_4$] = 1.0 μM). **e** Fluorescence response patterns of the sensor array against various proteins (23.8 μg/mL for each protein). **f** Canonical score plot for the two factors of simplified fluorescence response patterns obtained from LDA with 95% confidence ellipses. Error bars in **c**, **e** represent mean ± s.d. (*n* = 6 random numbers and independent experiments). The order of colored symbols from top to bottom in the legend corresponds to the column order of histogram from left to right in **c**, **e**.

Supplementary Table 10 and Supplementary Fig. 33, LDA analysis indicated that the three analytes could be identified and distinguished.

Experimentally, we selected the GC5A-CD/AlPcS$_4$ reporter pair and adjusted its ratio to create a new sensor array **SA7** (Fig. 6d). The concentration of AlPcS$_4$ was fixed at 1.0 μM, and the ratio of GC5A-CD to AlPcS$_4$ was 4:5, 1:1, 6:5, and 7:5 to construct the supramolecular sensor array. As shown in Fig. 6e, GC5A-CD had different affinities for various proteins, resulting

in distinguishable fluorescence signals in each sensor unit. From another point of view, the signals of different sensor units in response to one protein also caused visual differences. The obtained fluorescence change was used for LDA analysis, and 8 of the 13 proteins were discriminated with a 100% classification (Fig. 6f). One more sensor array based on adjusting reporter pair ratio was constructed with SC5A-CD and LCG in ratios of 4:5, 1:1, 6:5, and 7:5 (**SA8**). As shown in Supplementary Fig. 34, **SA8** was able to identify the 10 proteins.

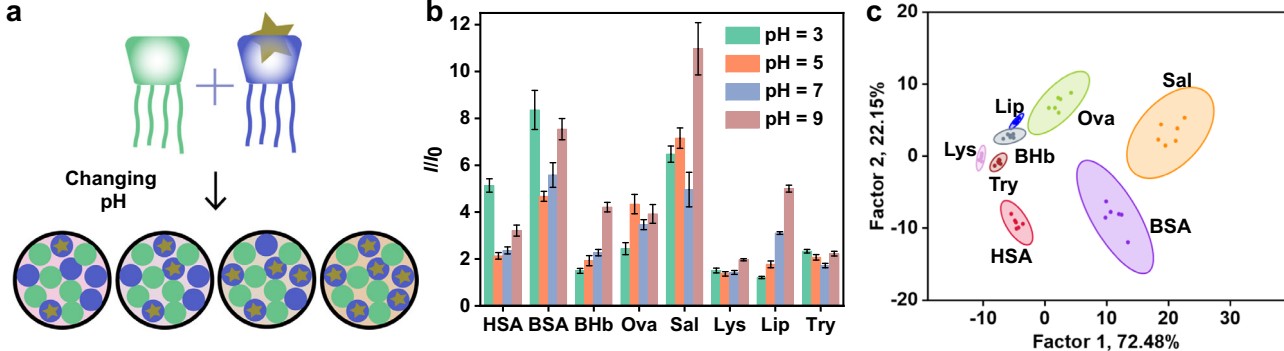

**Fig. 7 Construction and experimental results of sensor array based on changing the environmental factor. a** Schematic diagram of **SA9**. Pattern recognition of proteins using **SA9** ([GC5A] = [CD] = 1.0 μM, [AlPcS$_4$] = 1.0 μM). **b** Fluorescence response patterns of the sensor array against various proteins (23.8 μg/mL for each protein). The order of colored symbols from top to bottom in the legend corresponds to the column order of histogram from left to right. **c** Canonical score plot for the two factors of simplified fluorescence response patterns obtained from LDA with 95% confidence ellipses. Error bars in **b** represent mean ± s.d. ($n$ = 6 independent experiments).

**Sensor array based on changing the environmental factor.** Besides varying the molecular compositions, the sensor units constituting the array could also be adjusted by changing the environment, such as the polarity, viscosity, and pH. The binding affinities of a host-guest system are often influenced by the environment[60]. Taking advantage of the environmental sensitivity of supramolecular sensor arrays, different sensor units could be developed by using a single reporter pair at a fixed ratio and adjusting the external environment (Fig. 7a). The distinguishing ability comes from different recognition properties in the changing environments. First, we used a model to explain the underlying mechanism. The cross-reactivity of the sensor array is reflected in: (1) at a given pH, various analytes display different affinities to a single receptor and (2) at the three proposed pH values (Supplementary Table 11), the affinities among the same receptor, dye, and analyte are not exactly the same. When the simulated fluorescence signals were analyzed on the LDA plot, there was a clear distinguishing effect (Supplementary Table 12 and Supplementary Fig. 35).

As a proof of concept, the GC5A-CD/AlPcS$_4$ reporter pair was chosen as the sensor unit in aqueous solutions with pH = 3, 5, 7, and 9 to form a four-element sensor array (**SA9**). The model analytes here were proteins having different isoelectric points owing to their amino acid compositions. The pH affects not only the surface charges but also the conformations of proteins[68], potentially affecting their affinity toward CA-CD. Figure 7b shows the fluorescence response of each sensor unit to the proteins. As expected, the sensor units in different pH environments provided a unique response pattern for each protein. The LDA plot showed that among the 13 proteins, 8 were classified with 100% accuracy (Fig. 7c). To prove the universality of the pH adjustment method, another negatively charged coassembly, SC5A-CD, was complexed to LCG to construct the array (**SA10**). As shown in Supplementary Fig. 36, the eight proteins were successfully distinguished.

Table 1 summarizes the ten sensor arrays investigated in this study. All of them were able to discriminate the proteins. This is because the working principle of each sensor array is well-designed, which guarantees the discriminative ability in theory. The multiple interactions between receptor/dye, receptor/analyte, and dye/analyte, driven by hydrogen bond, electrostatic, hydrophobic, π−stacking, cation−π, and so on[69–71], formed the discriminative basis of these supramolecular sensor arrays. Additionally, the nonlinear relationship between fluorescence change and analyte concentration contributed also to the discrimination. The protein types discriminated by different sensor arrays were not exactly the

same. Therefore, we can combine two or more sensor arrays to obtain even better discriminative outcomes. The numbers of proteins that can be discriminated by combining two individual sensor arrays were shown in Supplementary Table 13. Among the 45 combinations, 21 combinations showed better results than the original two sensor arrays (Supplementary Figs. 37–39, Supplementary Table 13). The combination of **SA3** and **SA8** could discriminate all 13 proteins, which was not obtained by every single sensor array (Supplementary Fig. 38g). These combined sensor arrays also demonstrate the necessity to build more sensor units. Accordingly, one can form the sensor array by combining different construction methods, such as tuning both the coassembly and recognition ratios, to obtain a further refined discrimination index.

Although IDA is a powerful supramolecular sensing strategy, it still faces some limitations[6]. For instance, a single IDA sensing system is difficult to distinguish between low concentrations of a high-affinity analyte and high concentrations of a low-affinity analyte. Differential sensing based on IDA is reasonable to solve this problem. According to the simulation, the composite responses of the sensor array could form the fingerprint to two analytes in different concentrations, by assuming the binding affinities of different receptors to one analyte are not the same (Supplementary Tables 14, 15, Supplementary Fig. 40). We further employed **SA1** to valid the simulation result. Two proteins (CytC and Ova) with different concentrations were classified with 100% accuracy (Supplementary Fig. 41). This means even the concentrations and binding affinities of two (or more) analytes are unknown, the supramolecular sensor array can successfully distinguish them. The working principle is based on the binding difference between receptors and analytes. We also applied **SA1** to distinguish protein mixtures. Two blood proteins, BSA and BHb, were mixed with different percentages to be used as analyte samples. As shown in Supplementary Fig. 42, two pure proteins and two protein mixtures were classified by conducting LDA into four distinct clusters.

**Discrimination of actual complex systems.** We have developed supramolecular sensor arrays and constructed a variety of sensor units by simply changing the components, the ratio between the components, and the environment. These sensor arrays showed good discriminative abilities for proteins and their mixtures. Next, we wanted to determine whether they can function in more complex sample systems. Honey is a widely consumed natural food. It has a complex composition (mainly sugars, plus some enzymes, amino acids, vitamins, minerals, and aromatic substances)[72], but the honey types are difficult to discriminate because the texture, appearance, and smell of honey samples are very similar[73–75].

**Table 1 Summary of the ten supramolecular sensor arrays.**

| Sensor array | No. of receptors | No. of dyes | No. of pH values | No. of discriminated proteins | Working principle |
|---|---|---|---|---|---|
| SA1 | 5 (4CA+CD) | 2 | 1 | 11 | Different binding between receptor/dye, receptor/analyte, and dye/analyte |
| SA2 | 5 (4CA+CD) | 1 | 1 | 9 | Different binding between receptor/dye and receptor/analyte |
| SA3 | 2 (CA+CD) | 4 | 1 | 10 | Different binding between receptor/dye and dye/analyte |
| SA4 | 2 (CA+CD) | 4 | 1 | 9 | Different binding between receptor/dye and dye/analyte |
| SA5 | 2 (CA+CD) | 1 | 1 | 8 | Different binding between receptor/analyte |
| SA6 | 2 (CA+CD) | 1 | 1 | 9 | Different binding between receptor/analyte |
| SA7 | 2 (CA+CD) | 1 | 1 | 8 | Nonlinear relationship between fluorescence change and analyte concentration |
| SA8 | 2 (CA+CD) | 1 | 1 | 10 | Nonlinear relationship between fluorescence change and analyte concentration |
| SA9 | 2 (CA+CD) | 1 | 4 | 8 | Different binding between receptor/dye, receptor/analyte, and dye/analyte |
| SA10 | 2 (CA+CD) | 1 | 4 | 8 | Different binding between receptor/dye, receptor/analyte, and dye/analyte |

Therefore, we chose honey to test the ability of the supramolecular sensor arrays to discriminate real complex systems.

First, we tried to discriminate honey of the same brand but from different floral origins, because mislabeled flora origin is a frequent quality problem in the market. We selected four coassemblies (SC4A-CD, SC5A-CD, SC6A-CD, and SC8A-CD) and LCG as the reporter dye to form a four-element supramolecular sensor array (**SA11**). Acacia, linden, wolfberry, jujube, and vitex honey from the Tongrentang brand were used as the analytes. The four coassemblies can form host-guest complexes with one or more substances in the samples, but the binding affinities were different among the coassemblies due to the different cavity sizes of SCnAs. As shown in Fig. 8a, b, **SA11** was responsive to compounds in the honey samples and successfully classified samples of different floral origins. To verify the versatility of the sensor array in discriminating complex systems, we also tested jujube, acacia, motherwort, coptis, and multifloral honey from the Wangshi brand (Fig. 8c, d). The LDA plot showed very distinctive separation between samples of different floral origins. Considering that fructose, glucose, water, and maltose account for more than 93% of the weight in honey, these results prove that our sensor array can detect small differences in complex systems.

Next, we explored whether the sensor array could distinguish honey samples of the same floral origin but different brands. These honey samples are not exactly the same, because each company has its own production areas and processes. Further, since different brands command different market prices, a method to identify them has practical value. As shown in Fig. 8e, f, we applied the sensor array based on the SCnA-CD coassemblies to four commercially available brands of jujube honey. Although the samples have the same floral origin, they still elicited different responses. The difference in the fluorescence fingerprint generated by the four-element sensor array was sufficient to allow perfect distinction of the samples on the LDA plot. It can be said that this sensor array acts like a super-sensitive tongue that can taste and identify different honey samples.

The next challenge was to determine whether the sensor array could detect adulterated honey. We mixed genuine motherwort honey (Wangshi brand) with commercially available sirup in different proportions to simulate adulterated samples, and compared them with pure honey. As shown in Fig. 9a, even small amounts of added sirup changed the fluorescence response of the sensor array. Through LDA analysis, the two-dimensional score plot with 95% confidence ellipses shows that the six clusters for honey containing 0, 20%, 40%, 60%, 80%, and 100% sirup are clearly distinguished from each other (Fig. 9b). The clusters of pure honey and pure sirup are obviously far away. In the LDA plot, the first dominant factor accounts for 98.62% of the total variance, and the distribution of factor 1 varies with the percentage of sirup.

Finally, we conducted a more challenging experiment by mixing expensive honey (from vitex) with a cheap one (from rapeseed, at 0, 40%, 80%, and 100%). The host–guest pair of the sensor units can be incubated with the target mixtures to produce rapid and unique responses. As shown in Fig. 9c, the fluorescence responses to these samples were completely different. When the fluorescence responses were processed using LDA (Fig. 9d), these samples were well separated on the LDA plot with 100% classification accuracy. These findings show that the proposed supramolecular sensor arrays are useful for the discrimination and quality control of complex analytes such as commercial food products.

## Discussion
We have proposed a supramolecular approach to build a sensor array library by fully utilizing the recognition and assembly

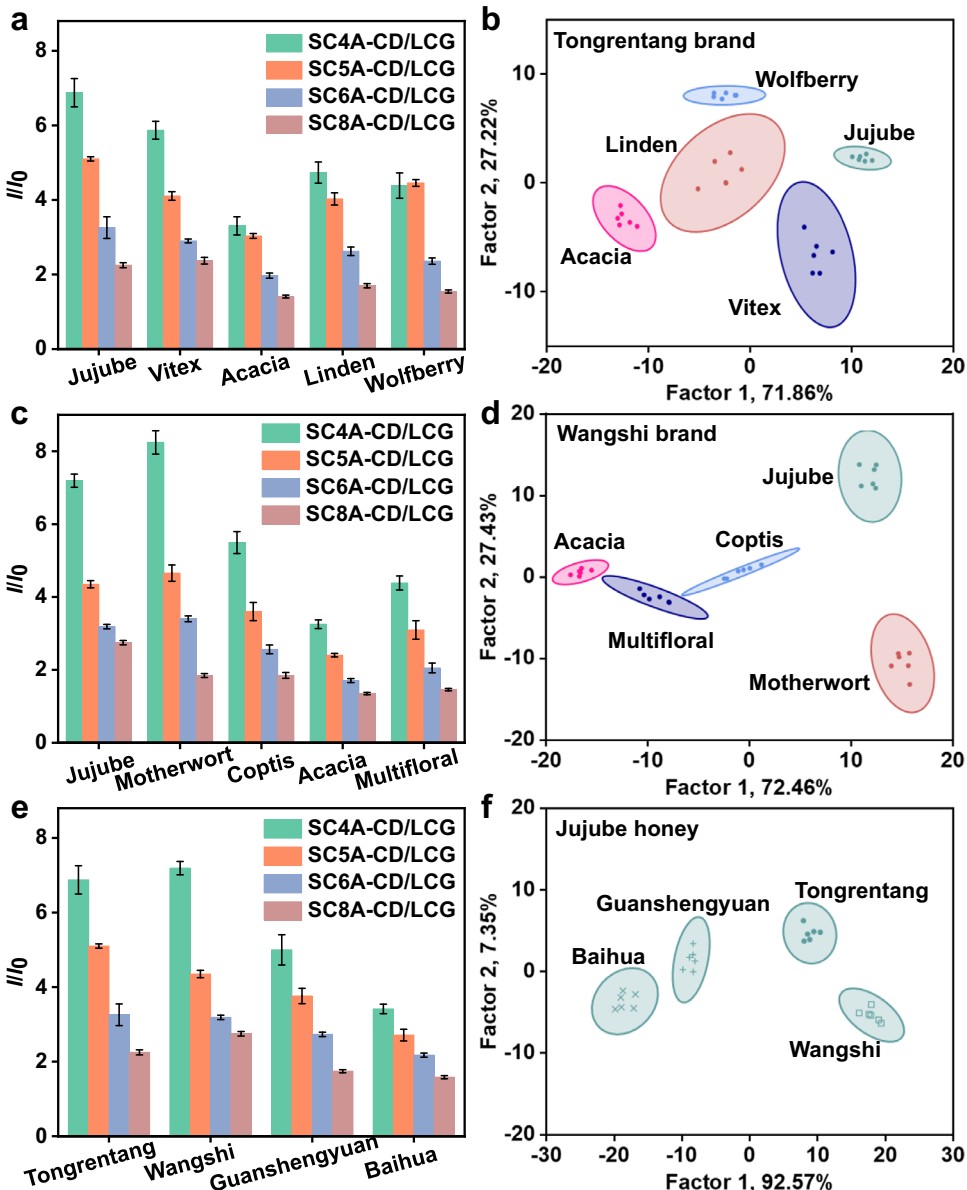

**Fig. 8 Discrimination of honey samples from different flora origins and brands. a** Fluorescence response patterns and **b** canonical score plot for Tongrentang brand samples. **c** Fluorescence response patterns and **d** canonical score plot for Wangshi brand samples. **e** Fluorescence response patterns and **f** canonical score plot for jujube honey samples (3.0 mg/mL for each honey sample). Pattern recognition of honey samples using **SA11** ([CA] = [CD] = 1.0 μM, [LCG] = 1.0 μM). Canonical score plots for the two factors of simplified fluorescence response patterns obtained from LDA with 95% confidence ellipses. Error bars in **a**, **c**, **e** represent mean ± s.d. (*n* = 6 independent experiments). The order of colored symbols from top to bottom in the legend corresponds to the column order of histogram from left to right in **a**, **c**, **e**.

capacity of macrocyclic amphiphiles. Upon replacing the components of the sensor units, adjusting the macrocycle/macrocycle coassembly ratio and the receptor/dye complexation ratio, and changing the environment, 32 sensor units were constructed and used as ten supramolecular sensor arrays. The discriminative abilities of these sensor units come from either (1) affinities between the receptors and analytes, the receptors and dyes, and additionally, the dyes and analytes or (2) the nonlinear relationship between the fluorescence signal and the analyte concentrations. Using proteins as model analytes, these ten sensor arrays showed effective discriminative abilities, and their discrimination as somewhat complementary. Accordingly, a sensor array could be constructed by combining different strategies to obtain a more refined discrimination index. We further applied this supramolecular sensor array strategy to analyze honey as a representative complex system. The sensor array showed the ability to discriminate honey samples from different floral origins and brands, as well as honey adulterated using sirup or cheaper honey. Owing to the dynamics of noncovalent interactions, this supramolecular strategy is a robust approach to constitutionally construct a sensor array library by simply switching the components or their ratios, which gives us chances to obtain better discriminative ability while using more limited building blocks. It also provides ideas for differential sensing by improving the richness of sensor array libraries with easy extension to other building blocks.

## Methods

**Materials**. All the reagents and solvents were commercially available and used as received unless otherwise specified purification. Iodomethane, ammonium hexafluorophosphate (NH$_4$PF$_6$), and tetrabutyl ammonium chloride hydrate (($n$-butyl)$_4$

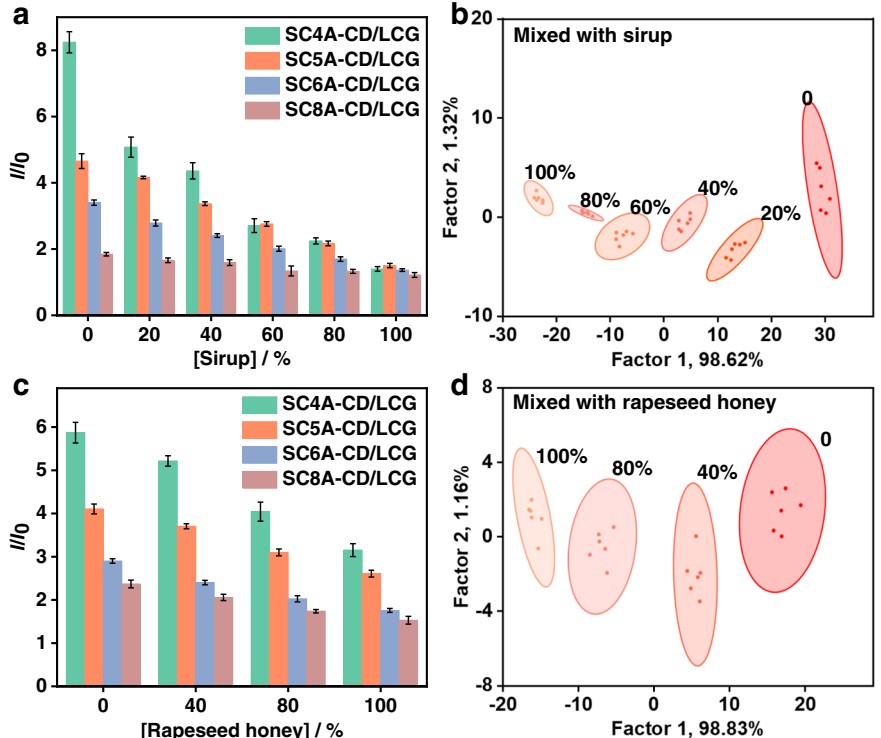

**Fig. 9 Discrimination of honey samples mixed with sirup or cheaper honey. a** Fluorescence response patterns and **b** canonical score plot against mixtures of honey and sirup. **c** Fluorescence response patterns and **d** canonical score plot against mixtures of honey and sirup against mixtures of vitex and rapeseed honey (3.0 mg/mL for each mixture sample). Pattern recognition of honey samples using **SA11** ([CA] = [CD] = 1.0 µM, [LCG] = 1.0 µM). Canonical score plots for the two factors of simplified fluorescence response patterns obtained from LDA with 95% confidence ellipses. Error bars in **a**, **c** represent mean ± s.d. (*n* = 6 independent experiments). The order of colored symbols from top to bottom in the legend corresponds to the column order of histogram from left to right in **a**, **c**.

NCl) were obtained from Sigma-Aldrich Co., Ltd. HSA, TRF, CytC and Mb were purchased from Shanghai Yuanye Bio-Technology Co., Ltd. BHb, Lip, and Lys were purchased from Shanghai Macklin Biochemical Co., Ltd. BSA and LCG were purchased from J&K Chemical. Sal was purchased from Dalian Meilun Bio-technology co., Ltd. Ova, Try, Pap, and sodium perchlorate (NaClO$_4$) were obtained from Aladdin. Human Hb was obtained from Sigma-Aldrich Co., Ltd. Nitric acid (HNO$_3$) and sodium hydroxide (NaOH) were purchased from Tianjin Fengchuan Chemical Reagent Co., Ltd. AlPcS$_4$ was obtained from Frontier Sci-entific. TPPS, 1,8-ANS, and 2,6-TNS were purchased from Tokyo Chemical Industry. PTPE was prepared according to the previous literature procedure (Supplementary Figs. 10, 23)[76]. The protein samples were dissolved at 5.0 mg/mL in water as stock solutions. Then the protein stock solution was diluted to the corresponding low concentration in subsequent sensing experiments. The honey samples were all commercially available brands in the Chinese market and they were dissolved at 50.0 mg/mL in water and stored at 4 °C until analysis. All solu-tions were prepared using ultrapure water from Thermo Scientific purification system. Different pH solutions were prepared by titration with NaOH for pH 7 and 9 at 25 °C, and titration with HNO$_3$ for pH 3 and 5 at 25 °C.

**Instrumentation**. NMR data were recorded on a Bruker AV400 spectrometer and Zhongke-Niujin BIXI-I 400 spectrometer. Mass spectra were recorded on an Agilent 6520 Q-TOF LC/MS. Elemental analysis measurements were performed by Elementar Vario EL Cube. Melting points were measured by Yuhua X-4 micro-scopic melting point apparatus. The pH values were measured by Mettler Toledo FiveEasy Plus. The dynamic light scattering and zeta potential were examined on a NanoBrook 173plus laser light scattering spectrometer equipped with a digital correlator at 659 nm (scattering angle of 90°). Steady-state fluorescence spectra were recorded in a conventional quartz cell (light path 10 mm) on an Agilent Cary Eclipse spectrometer equipped with a Cary single-cell Peltier accessory or a Cary Eclipse microplate reader accessory.

**Discriminant analysis**. 200 µL of CA-CD coassemblies and corresponding dyes were added to a black 96-well plate and incubated at 25 °C for 10 min. The specific concentrations of sensor arrays were indicated in the figure legend. The excitation wavelengths for AlPcS$_4$, LCG, TPPS, 1,8-ANS, PTPE, and 2,6-TNS were 610, 365, 412, 350, 327, and 350 nm, respectively. The initial fluorescence intensities at the maximum emission wavelengths of dyes were recorded as $I_0$. The maximum

emission wavelengths for AlPcS$_4$, LCG, TPPS, 1,8-ANS, PTPE, and 2,6-TNS were 680, 500, 650, 460, 479, and 422 nm, respectively. The target analytes (10 µL proteins at 0.5 mg/mL, or 13 µL honey at 50 mg/mL) were introduced to each well and incubated for 10 min. Then the fluorescence intensity at maximum emission wavelength of each well was recorded as $I$. The obtained relative fluorescence intensities ($I/I_0$) were used as the response signals for array sensing analysis. Each experiment was repeated six times. Finally, the raw data matrix was handled using LDA in Past 3 program. The full score plots of 13 proteins were shown in Sup-plementary Fig. 43.

## Data availability
Data supporting the findings of this study are available within the paper and its Supplementary Information. The source data underlying Figs. 3–9, and Supplementary Figs. 24–36, 40–42 are provided as a Source Data file. Additional data are available from the corresponding author upon request. Source data are provided with this paper.

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

## Acknowledgements

This work was supported by NSFC (22101142 and U20A20259), the Fundamental Research Funds for the Central Universities, and the NCC Fund (grant no. NCC2020FH04), which are gratefully acknowledged. The authors also thank Prof. Bart Jan Ravoo at Westfälische Wilhelms-Universität Münster for supplying the amphiphilic cyclodextrin used in this work.

## Author contributions

J.H.T. and X.Y.H. contributed equally. D.S.G. and X.Y.H. devised the project. J.H.T. carried out the experimental work and analyzed the data. Z.Y.H., H.W.T., J.J.L., and H.B.L. synthesized the compounds. Y.C.P. helped to prepare the coassemblies. J.H.T., X.Y.H., and D.S.G. wrote the manuscript.

## Competing interests

The authors declare no competing interests.
