## [Peer Review File · Nature Communications]

A Facile Way to Construct Sensor Array Library via Supramolecular Chemistry for Discriminating Complex SystemsREVIEWER COMMENTS

Reviewer #1 (Remarks to the Author):

This manuscript describes a general route to the production of differential sensor arrays by adopting a combinatorial approach to established indicator displacement assay (IDA) methodology. The approach is novel, and the performance of the sensor arrays in discriminating proteins is impressive. The system has been conceptualised for differentiation of proteins, yet enables differentiation of complex mixtures like honey, where the protein concentration is low, which is remarkable. I have, however, some technical and other questions/comments that must be addressed before I could recommend publication of this manuscript.

1. The design principles of the system are not entirely clear to me. This is not a major problem, as many sensor arrays are hypothesis-free in their design, but the authors reference interactions between receptors and analytes, and dyes and analytes. Could the authors please comment on the nature of these interactions?
2. It would be helpful if the initial discussion of design methodology could explain the basic principles of analyte detection via IDAs and refer interested readers to a review (around p4, line 52 onwards). This early explanation would make the description of the authors' system easier to follow.
3. Scheme 1: stereochemistry at the anomeric position of cyclodextrin is not defined. The text should specify that β -cyclodextrin is used. In the structure of GCnAs, the positive charge is displayed on H rather than N. Inclusion of counterions in the scheme would be helpful.
4. The manuscript does not address a key limitation of IDA methodology – how can differences in analyte concentration be accounted for? How can the array distinguish between low concentrations of a high affinity analyte and high concentrations of a low affinity analyte? The robustness of the sensor array in this respect could be established by exposing the array to varying concentrations of a single analyte, and using the original training set to identify the analyte.
5. I would question the benefit of presenting simulated data in the main body of the manuscript – these plots could be moved to the SI. In particular, the inclusion of simulated fluorescence spectra could confuse the reader.
6. Conditions such as pH and ionic strength should be noted in captions. I note from the methods section that solutions of analytes were not buffered. Is dye emission likely to be affected by changes in pH/ionic strength of surrounding medium? Some results presented (p17-19) suggest so, and this factor could limit the ability of the sensor array to discriminate 'unknown' analytes where these conditions are not known/controlled.
7. Methods section (p26, line 461) suggests that protein solutions have been made up at a concentration of 5.0 mg/mL while the main text states that solutions were made at 23.8 μ g/mL. Could the authors please clarify the concentration used?
8. p29, line 475: Could authors specify the temperature at which the experiment was conducted?
9. Characterisation data for synthesised compounds lacks an indication of purity e.g. melting point, elemental analysis, HPLC.
10. DLS data: it is unclear if the particle size distributions presented are intensity, number or volume distributions. This detail should be stated, and the correlation functions for the measurements should be shown also.
11. SI, p9: equation to denote HD/HC complexation contain merged/superimposed characters.

12. p13, line 239: the word “fluorescent” should be changed to “fluorescence.”

Reviewer #2 (Remarks to the Author):

This manuscript describes the assembly of a series of supramolecular indicator displacement arrays (IDAs). The sensor arrays were able to discriminate sample proteins and also honey. The authors highlight the efficiency in constructing the individual discriminating sensor units using self-assembly. However, this is not new as Anslyn developed indicator displacement arrays as early as 1999 based on self-assembling synthetic receptors and dyes and there are now many examples of this strategy being applied to various sensing and sample discrimination applications.

Perhaps the more interesting aspect of this work is the systematic study of the individual variables that can be changed to create potentially complementary sensor arrays. While these individual variables such as dyes, synthetic receptors, concentrations, ratios, and environment, have all been used to create complexity in sensor arrays, this reviewer does not know any example where these variables have been systematically studied and compared within a single system. Unfortunately, this systematic study is flawed or incomplete. For example in the truncated sensor arrays (SA2 through SA7), the authors only examined a subset of the variable space and thus, their conclusions about a particular variable being effective or comparing the effectiveness of variables is cannot be made with any certainty. For example, in SA3 where the dyes were varied and the CD/CA complex was kept constant, only one CD/CA combination was studied (CD/GC5A). While good differentiation was observed in discriminating 9 of the 13 protein samples, it is unclear whether CD/GC5A was picked randomly or was the best assembly for this variable, and therefore the conclusions and comparisons of the dye variable are limited. The same is true for the other variables examined in Table 1 for sensor arrays SA3-S7. Even the conclusion that combining two or more variables to create a more discriminating sensor array (while logical) cannot be proven using the data presented as it is shown to be an effective approach in one example (SI Figure 15) but we do not know if this example is the exception or the rule.

Also problematic was the shift in manuscript from the development to application. All of the development work was done by testing protein samples but the applications were performed using honey samples, which are largely sugars and carbohydrates. While it is likely that the general design principles that were developed in the protein sensor arrays can be applied to the honey samples, one cannot definitively support this conclusion from the data presented.

Given the issues above, this manuscript is not recommended for publication in Nature Communications.

Some additional comments and questions

What is the role of the cyclodextrins in the supramolecular assemblies? All of the figures show the dyes bound to the calixarenes and not the cyclodextrins. Do the cyclodextrins also compete for the dyes and quench their fluorescence?

In Figure 3b, the figure caption and the manuscript text states that the CD/CA combination was kept constant but the legend for Figure 3b shows four different CD/CA combinations.

The discussion of the origins of the effectiveness of the concentration variable arising from the non-linear binding curves was a very insightful.

Reviewer #3 (Remarks to the Author):

The submission by Tian et al. reports a new concept in the construction and use of chemical sensor arrays. Where previous sensor arrays are built by synthesis of individual analyte-binding sensor elements, the new approach here is to build sensor arrays by unique combinations of a small number of sensor elements that have both the ability to bind analytes and also the ability to co-assemble into aggregated multicomponent sensors. It was questionable whether such aggregate combinations would provide effective differences in analyte responses. That is, whether the responses would be different enough to be useful in identifying and quantifying different analytes. The answer, proved in a series of systematic studies, is definitely yes. Rather than go into each piece of experimental evidence in detail, I'll report for the editors that the authors have been incredibly systematic in exploring how such co-assembled sensor arrays perform. They explore sensor arrays in which both hosts and dyes are varied, then arrays in which the dye is held constant, then arrays in which the co-assembly is held constant and the dye is varied, then arrays in which only the co-assembly ratio is varied, then arrays in which the host-dye ratio is varied, then arrays in which the solution pH is varied. Each of these variations is operationally simple, and each demonstrates the ability to discriminate many different analytes. By exploring mix-and-match array construction in such a systematic way, the authors arrive at a series of conclusions that go far beyond the typical sensor array paper. Where most sensor array papers focus on "can it discriminate between these analytes?" in a very limited case, this paper provides a theoretical basis for building sensor arrays in a conceptually new way that will have enormous, sweeping impact on the field.

I have one request for a minor edit. Each sensor array is able to discriminate between many analytes, but some analytes end up failing to be separated into distinct regions of the scores plots and are then omitted. Please provide all full scores plots, including data for the analytes that are not successfully discriminated, in the supporting information. While those data are not central to the case the authors are making, they might provide interesting material for specialists who want to understand these systems in more details.

REVIEWER COMMENTS

Reviewer #1 (Remarks to the Author):

This manuscript describes a general route to the production of differential sensor arrays by adopting a combinatorial approach to established indicator displacement assay (IDA) methodology. The approach is novel, and the performance of the sensor arrays in discriminating proteins is impressive. The system has been conceptualised for differentiation of proteins, yet enables differentiation of complex mixtures like honey, where the protein concentration is low, which is remarkable. I have, however, some technical and other questions/comments that must be addressed before I could recommend publication of this manuscript.

Response: We highly appreciate the reviewer's positive and constructive comments.

1. The design principles of the system are not entirely clear to me. This is not a major problem, as many sensor arrays are hypothesis-free in their design, but the authors reference interactions between receptors and analytes, and dyes and analytes. Could the authors please comment on the nature of these interactions?

Response: Thanks for the kind advice. For the nature of these interactions, we added the description "The multiple interactions between receptor/dye, receptor/analyte, and dye/analyte, driven by hydrogen bond, electrostatic, hydrophobic, π -stacking, cation- π , and so on,^[67-69] formed the discriminative basis of these supramolecular sensor arrays. Additionally, the nonlinear relationship between fluorescence change and analyte concentration contributed also to the discrimination." in the revised manuscript (Page 20) and highlighted it.

2. It would be helpful if the initial discussion of design methodology could explain the basic principles of analyte detection via IDAs and refer interested readers to a review (around p4, line 52 onwards). This early explanation would make the description of the authors' system easier to follow.

Response: We believe an early explanation of IDA in the introduction will help the readers understand our design principle. According to the reviewer's advice, we added the description "For instance, the indicator displacement assay (IDA), in which a competitive analyte is introduced to a dye/receptor system and the displacement of dye by analyte modulates an optical signal, represents a popular supramolecular strategy to construct sensor array." in the revised manuscript (Page 4) and referred two reviews about IDA.

3. Scheme 1: stereochemistry at the anomeric position of cyclodextrin is not defined. The text should specify that β -cyclodextrin is used. In the structure of GCnAs, the positive charge is displayed on H rather than N. Inclusion of counterions in the scheme would be helpful.

Response: Thanks for the kind advice. We have defined the stereochemistry of cyclodextrin in Scheme 1. The position of the positive charge has been corrected and the counterions were also added in Scheme 1. We also classified the β -CD in the sentence: "Here, we chose nine different macrocycles, namely guanidinium calix[n]arenes (GCnAs, $n = 4$ or 5), quaternary ammonium calix[n]arenes (QCnAs, $n = 4$ or 5), sulfonated calix[n]arenes (SCnAs, $n = 4, 5, 6, 8$) and amphiphilic β -CD (Scheme 1, Supplementary Scheme 1 and Supplementary Figs. 1-6) as the building blocks of coassembled receptors, because of their excellent abilities to recognize different guest molecules." But for concise, we still used CD when we mentioned the coassembly such as GC5A-CD.

4. The manuscript does not address a key limitation of IDA methodology – how can differences in analyte concentration be accounted for? How can the array distinguish between low concentrations of a high affinity analyte and high concentrations of a low affinity analyte? The robustness of the sensor array in this respect could be established by exposing the array to varying concentrations of a single analyte, and using the original training set to identify the analyte.

Response: Thanks for the kind advice. As the reviewer mentioned, sensor array is able to address this key limitation of IDA, which was ignored in our original manuscript. Accordingly, we added a simulation to prove the feasibility that sensor array can address this limitation of IDA in theory, and then applied a supramolecular sensor array (**SA1**) to distinguish two proteins (CytC and Ova) in different concentrations (2.4, 3.6, 4.8, 14.3, 71.4, 238.1 $\mu\text{g/mL}$). The different concentrations of these two proteins were clearly distinguished. The simulation and experimental data were shown in Supplementary Tables 14–15, Supplementary Figs. 23–24 and described in revised manuscript (Page 22).

5. I would question the benefit of presenting simulated data in the main body of the manuscript – these plots could be moved to the SI. In particular, the inclusion of simulated fluorescence spectra could confuse the reader.

Response: According to the reviewer's suggestion, we moved the simulated fluorescence spectra in Fig. 2 to SI (Supplementary Fig. 10) to avoid the confusion. However, in Fig. 5, the explanation for the non-linear binding curves is necessary for the discussion, and the simulated curves are helpful for understanding, especially for readers who are not familiar with molecular recognition. In order to make the manuscript more reader friendly, the simulated curves and fluorescence spectra in Fig. 5 were still kept in the revised manuscript.

6. Conditions such as pH and ionic strength should be noted in captions. I note from the methods section that solutions of analytes were not buffered. Is dye emission likely to be affected by changes in pH/ionic strength of surrounding medium? Some results presented (p17-19) suggest so, and this factor could limit the ability of the sensor array to discriminate 'unknown' analytes where these conditions are not known/controlled.

Response: pH and ionic strength would affect the binding affinities between receptor/dye, receptor/analyte and analyte/dye, therefore, we can use the change of pH to construct sensor arrays (**SA9** and **SA10** in this revised manuscript), and the pH values were controlled although we did the experiments in water.

For the other sensor arrays, we investigated the effect of pH and ionic strength to dye emission. In our experimental conditions, after adding the experimental concentration of proteins, the pH values were only slightly changed (around 6.5), therefore, during the experiment, the pH was stable and did not affect the emission of dyes (Supplementary Table 2). For the ionic strength, we tested the spectra of dyes we used in this work in the presence of NaClO_4 (up to 27.5 mM), and the fluorescence spectra were only slightly changed (Supplementary Fig. 9). Therefore, we believed in our experimental condition, the pH was stable and ionic strength of surrounding medium did not influence the dye emission seriously. We have revised the caption of Fig. 2 and clarified the effect of pH and ionic strength.

7. Methods section (p26, line 461) suggests that protein solutions have been made up at a concentration of 5.0 mg/mL while the main text states that solutions were made at 23.8 $\mu\text{g/mL}$. Could the authors please clarify the concentration used?

Response: 5.0 mg/mL is the concentration of the stock concentration of proteins. 23.8 µg/mL is the concentration we used in the discrimination experiments. To make the description clearer, we changed it as “*The protein samples were dissolved at 5.0 mg/mL in water as stock solutions. Then the protein stock solution was diluted to the corresponding low concentration in subsequent sensing experiments.*” in revised manuscript (Page 28).

8. p29, line 475: Could authors specify the temperature at which the experiment was conducted?

Response: We changed the description “*200 µL of CA-CD coassemblies and corresponding dyes were added to a black 96-well plate and incubated at 25 °C for 10 minutes.*” in the revised manuscript (Page 29).

9. Characterisation data for synthesised compounds lacks an indication of purity e.g. melting point, elemental analysis, HPLC.

Response: According to the reviewer’s advice, we have added the data of melting point and elemental analysis in supporting information (Supplementary Note 1). Unfortunately, we did not obtain the HPLC result because the polarity of calixarenes is so high that we haven’t found the HPLC conditions that suitable to monitoring these calixarenes.

10. DLS data: it is unclear if the particle size distributions presented are intensity, number or volume distributions. This detail should be stated, and the correlation functions for the measurements should be shown also.

Response: We refined the DLS data plot, that the particle size distributions were number distributions (Supplementary Fig. 7). Additionally, we have added the correlation functions for the measurements (Supplementary Fig. 7).

11. SI, p9: equation to denote HD/HC complexation contain merged/superimposed characters.

Response: We sincerely thank the reviewer for the careful inspection and reminder. We have corrected it to the correct format in SI.

12. p13, line 239: the word “fluorescent” should be changed to “fluorescence.”

Response: We have changed the “fluorescent” to “fluorescence” in the revised manuscript.

Reviewer #2 (Remarks to the Author):

This manuscript describes the assembly of a series of supramolecular indicator displacement arrays (IDAs). The sensor arrays were able to discriminate sample proteins and also honey. The authors highlight the efficiency in constructing the individual discriminating sensor units using self-assembly. However, this is not new as Anslyn developed indicator displacement arrays as early as 1999 based on self-assembling synthetic receptors and dyes and there are now many examples of this strategy being applied to various sensing and sample discrimination applications.

Response: We highly appreciate the reviewer's comments. As mentioned by the reviewer, the concept of IDA has been proposed by Eric Anslyn in 1999 (Inouye and Shinkai have used this strategy a little earlier: Inouye, M.; Hashimoto, K.; Isagawa, K., *J. Am. Chem. Soc.*, **1994**, *116*, 5517–5518; Koh, K. N.; Araki, K.; Ikeda, A.; Otsuka, H.; Shinkai, S., *J. Am. Chem. Soc.*, **1996**, *118*, 755–758.), and has been applied to sensing various analytes including anions, chiral molecules, carbohydrates, and biomolecules. But IDA still has some problems, for instance, a single IDA system is difficult to determine low concentrations of a high affinity analyte and high concentrations of a low affinity analyte. Therefore, differential sensing was introduced (see response to Q4 by Reviewer 1 please). Because sensor units can be easily constructed by combining multiple receptors and multiple indicators through IDA strategy, many sensor arrays have been constructed by IDA approach using synthetic receptors and dyes. But these sensor arrays were developed by either self-assembly or recognition. The major innovation of our present work is proposing a new design principle for supramolecular sensor array. The subtle combination of assembly and recognition overcomes a main challenge in differential sensing: to build a library of sensor units. Using our strategy, numerous of sensor arrays can be constructed by limited sensor elements though replacing the components of the sensor units, adjusting the macrocycle/macrocycle coassembly ratios, the receptor/dye complexation ratios, and changing the environments. This is the charming point of supramolecular chemistry, which is difficult for synthetic chemistry. Moreover, this strategy is amenable to other supramolecular amphiphilic systems. The novelty of this work is providing a supramolecular approach for building sensor arrays with a theoretical basis, but does not rely on which analytes we discriminated or how much analytes we discriminated. Just like Reviewer 3 said "Where most sensor array papers focus on "can it discriminate between these analytes?" in a very limited case, this paper provides a theoretical basis for building sensor arrays in a conceptually new way that will have enormous, sweeping impact on the field."

Perhaps the more interesting aspect of this work is the systematic study of the individual variables that can be changed to create potentially complementary sensor arrays. While these individual variables such as dyes, synthetic receptors, concentrations, ratios, and environment, have all been used to create complexity in sensor arrays, this reviewer does not know any example where these variables have been systematically studied and compared within a single system. Unfortunately, this systematic study is flawed or incomplete. For example, in the truncated sensor arrays (SA2 through SA7), the authors only examined a subset of the variable space and thus, their conclusions about a particular variable being effective or comparing the effectiveness of variables is cannot be made with any certainty. For example, in SA3 where the dyes were varied and the CD/CA complex was kept constant, only one CD/CA combination was

studied (CD/GC5A). While good differentiation was observed in discriminating 9 of the 13 protein samples, it is unclear whether CD/GC5A was picked randomly or was the best assembly for this variable, and therefore the conclusions and comparisons of the dye variable are limited. The same is true for the other variables examined in Table 1 for sensor arrays SA3-S7. Even the conclusion that combining two or more variables to create a more discriminating sensor array (while logical) cannot be proven using the data presented as it is shown to be an effective approach in one example (SI Figure 15) but we do not know if this example is the exception or the rule.

Response: According to the reviewer's advice, to further prove the generality of our design principle, we performed more experiments and ensured that each building principle had two individual sensor arrays. The corresponding discussion of new sensor arrays (**SA4**: using QC5A-CD, **SA6**: using SC4A-CD and **SA8**: using SC5A-CD) were highlighted in the revised manuscript and revised supporting information (see in Page 14, 15, 18). These new sensor arrays all showed good discrimination abilities, which proved that our design principles were available to other CA-CD sensor units.

We also added the discriminating effect of combining every two sensor arrays. The results were shown in the revised manuscript and supporting information (manuscript: Page 20; supporting information: Supplementary Figs. 20–22, Supplementary Table 13). Among the 45 combinations, 21 combinations performed better than the two original one. It is needed to note that after the addition of new sensor arrays, we obtained the total discrimination of 13 proteins (**SA3** + **SA8**), which is better than the combination of our "old" sensor arrays. So it could be a general rule to obtain better results by combining two sensor arrays. This also further proved the necessity to build the sensor unit library, because the good discriminating effect is not only related to the recognition ability of single sensor unit, but also the combination of different sensor units. We added the description in the revised manuscript (Page 20).

Also problematic was the shift in manuscript from the development to application. All of the development work was done by testing protein samples but the applications were performed using honey samples, which are largely sugars and carbohydrates. While it is likely that the general design principles that were developed in the protein sensor arrays can be applied to the honey samples, one cannot definitively support this conclusion from the data presented.

Response: The main point of this work lies to the simple construction method of sensor units, instead of discriminating special analytes. The obtained supramolecular sensor arrays are capable of discriminating both protein and honey samples, demonstrating these arrays are powerful to be applied in a wide scope of analytes.

In order to make this shift more smoothly, we added an experiment that using **SA1** to discriminate the mixture of two proteins (BSA and BHp). The description was added in the revised manuscript (Page 22).

Given the issues above, this manuscript is not recommended for publication in Nature Communications.

Some additional comments and questions

What is the role of the cyclodextrins in the supramolecular assemblies? All of the figures show the dyes bound to the calixarenes and not the cyclodextrins. Do the cyclodextrins also compete

for the dyes and quench their fluorescence?

Response: The employed dyes were only bound to calixarenes (CAs), while cyclodextrin (CD) did not quench these dyes (Fig. R1). But CD is still important in these sensor units. CD and CAs can provide different binding sites, which show synergistic effect in recognition. In our previous work (Xu, Z.; Jia, S.; Wang, W.; Yuan, Z.; Ravoo, B.; Guo, D.-S., *Nat. Chem.*, **2019**, *11*, 86–93), we have systematically investigated and discussed the binding affinities of the CA-CD coassembly and the CA assembly to different model peptides. The binding constants were determined by fluorescent IDA titration (CA and lucigenin as the reporter pair). The results showed the CA-CD coassembly showed significantly stronger binding to some model peptides than the CA assembly, which proved that coassembling CA and CD indeed improved the binding efficiency and selectivity. Tuning the ratio of CA/CD would lead to distinguishable binding capabilities. The differential recognition is the basis for the sensor array.

Fig. R1. The fluorescence spectra of dyes before and after adding β -CD ($[\beta\text{-CD}] = 2.0 \mu\text{M}$, $[\text{dye}] = 1.0 \mu\text{M}$).

In Figure 3b, the figure caption and the manuscript text states that the CD/CA combination was kept constant but the legend for Figure 3b shows four different CD/CA combinations.

Response: We are sorry about the mistake, the data of Fig. 3b is right, but the legend is wrong. We have corrected this mistake in the revised manuscript.

The discussion of the origins of the effectiveness of the concentration variable arising from the non-linear binding curves was a very insightful.

Response: We thank the reviewer's positive comment on the point of the non-linear binding curve. The non-linear relationship between the fluorescence signal and analyte concentration in binding titrations is common in supramolecular chemistry but has not been applied as the principle for constructing sensor array. We believe that the non-linear binding curve would gain more and more attention for the development of differential sensing in the future.

Reviewer #3 (Remarks to the Author):

The submission by Tian et al. reports a new concept in the construction and use of chemical sensor arrays. Where previous sensor arrays are built by synthesis of individual analyte-binding sensor elements, the new approach here is to build sensor arrays by unique combinations of a small number of sensor elements that have both the ability to bind analytes and also the ability to co-assemble into aggregated multicomponent sensors. It was questionable whether such aggregate combinations would provide effective differences in analyte responses. That is, whether the responses would be different enough to be useful in identifying and quantifying different analytes. The answer, proved in a series of systematic studies, is definitely yes. Rather than go into each piece of experimental evidence in detail, I'll report for the editors that the authors have been incredibly systematic in exploring how such co-assembled sensor arrays perform. They explore sensor arrays in which both hosts and dyes are varied, then arrays in which the dye is held constant, then arrays in which the co-assembly is held constant and the dye is varied, then arrays in which only the co-assembly ratio is varied, then arrays in which the host-dye ratio is varied, then arrays in which the solution pH is varied. Each of these variations is operationally simple, and each demonstrates the ability to discriminate many different analytes. By exploring mix-and-match array construction in such a systematic way, the authors arrive at a series of conclusions that go far beyond the typical sensor array paper. Where most sensor array papers focus on "can it discriminate between these analytes?" in a very limited case, this paper provides a theoretical basis for building sensor arrays in a conceptually new way that will have enormous, sweeping impact on the field.

I have one request for a minor edit. Each sensor array is able to discriminate between many analytes, but some analytes end up failing to be separated into distinct regions of the scores plots and are then omitted. Please provide all full scores plots, including data for the analytes that are not successfully discriminated, in the supporting information. While those data are not central to the case the authors are making, they might provide interesting material for specialists who want to understand these systems in more details.

Response: We highly appreciate the reviewer's positive and constructive comments. The full scores plots were added in the supporting information (Supplementary Fig. 26).

REVIEWERS' COMMENTS

Reviewer #1 (Remarks to the Author):

I am happy with the amendments made to the manuscript, and am happy to recommend it for publication.

Reviewer #2 (Remarks to the Author):

The authors have addressed most of the concerns and issues raised by this reviewer especially those relating to the novelty of the work and whether the conclusions about the mechanism and applicability of the work could be supported by the data presented. Therefore, this manuscript is now recommended for publication in Nature Communications.

The authors have more clearly explained the novelty of their supramolecular sensor array system in the authors' responses. Specifically, how it is differentiated from previous receptor-dye arrays by generating new sensors by varying the dye and receptor ratios. However, it is not obvious that varying the receptor-dye ratios would yield individual sensors with unique responses that are required for a sensor array without considering the non-linear responses of recognition systems. Therefore, an explanation of how the non-linear responses of supramolecular systems allows different receptor-dye ratios to provide unique responses should be incorporated into the introduction. If the authors could include a scheme of how a non-linear response leads to different responses patterns for different receptor-dye ratios, it would be very helpful to the readers in understanding the underlying hypothesis of this work. Currently, there is only a brief reference to the non-linear response of supramolecular systems in the caption for Fig 1 in the introduction.

Reply to Reviewers

Reviewer #1 (Remarks to the Author):

I am happy with the amendments made to the manuscript, and am happy to recommend it for publication.

Response: We highly appreciate the reviewer for the recommendation for publication.

Reviewer #2 (Remarks to the Author):

The authors have addressed most of the concerns and issues raised by this reviewer especially those relating to the novelty of the work and whether the conclusions about the mechanism and applicability of the work could be supported by the data presented. Therefore, this manuscript is now recommended for publication in Nature Communications.

Response: We highly appreciate the reviewer for the recommendation for publication.

The authors have more clearly explained the novelty of their supramolecular sensor array system in the authors' responses. Specifically, how it is differentiated from previous receptor-dye arrays by generating new sensors by varying the dye and receptor ratios. However, it is not obvious that varying the receptor-dye ratios would yield individual sensors with unique responses that are required for a sensor array without considering the non-linear responses of recognition systems. Therefore, an explanation of how the non-linear responses of supramolecular systems allows different receptor-dye ratios to provide unique responses should be incorporated into the introduction. If the authors could include a scheme of how a non-linear response leads to different responses patterns for different receptor-dye ratios, it would be very helpful to the readers in understanding the underlying hypothesis of this work. Currently, there is only a brief reference to the non-linear response of supramolecular systems in the caption for Fig 1 in the introduction.

Response: We highly appreciate the reviewer's positive and constructive comments. As the reviewer mentioned in previous comment "the effectiveness of the concentration variable arising from the non-linear binding curves was very insightful", the nonlinear response is indeed an emerging factor for building sensor array, but we did not give enough emphasis about it in previous version. Therefore, according to the reviewer's suggestion, we added some description about the nonlinear response in introduction "Moreover, during titration, the nonlinear relationship between the output signal and the component concentration can be exploited to construct sensor units by simply adjusting the receptor/dye ratios. This nonlinear relationship has ever been used in supramolecular encryption²⁷⁻²⁸, but is overlooked in differential sensing. Such sensor units generate different signals for each analyte due to complex supramolecular equilibria even the binding affinities among receptor, dye and analyte stay the same." in the revised manuscript. About the scheme, we also think a suitable scheme could help the readers to understand this nonlinear principle. We tried to draw several schemes to show how the nonlinear relationship led to different response patterns by adjusting reporter pair ratios, but we did not find anyone is better than Fig. 6b. So we still use Fig. 6b to illustrate the nonlinear principle. We also give the brief description in the section of "Introduction" and detailed discussion about it in the section of "Sensor array based on adjusting reporter pair ratio". We hope the readers could understand this nonlinear principle clearly.